Synovial fluid proteome profile of surgical versus chemical induced osteoarthritis in rabbits

Syed Sulaiman Sharifah Zakiah 1
Tan Wei Miao 1
Radzi Rozanaliza 1
Shafie Intan Nur Fatiha 1
Ajat Mokrish 2
Mansor Rozaihan 3
Mohamed Suhaila 4
Rahmad Norasfaliza 5
Ng Angela Min Hwei 6
Lau Seng Fong lausengfong@upm.edu.my lausengfong@hotmail.com 1 4
1 Department of Veterinary Clinical Studies, Universiti Putra Malaysia , Serdang , Selangor , Malaysia
2 Department of Veterinary Preclinical Studies, Universiti Putra Malaysia , Serdang , Selangor , Malaysia
3 Department of Farm and Exotic Animals Medicine and Surgery, Universiti Putra Malaysia , Serdang , Selangor , Malaysia
4 Laboratory of Cancer Research UPM-MAKNA (CANRES), Institute of Bioscience, Universiti Putra Malaysia , Serdang , Selangor , Malaysia
5 Agro Biotechnology Institute , Serdang , Selangor , Malaysia
6 Tissue Engineering Centre, Universiti Kebangsaan Malaysia , Cheras , Kuala Lumpur , Malaysia
Uversky Vladimir
Electronic publication date: 2022 Feb 23
Publication date: 2022
Volume: 10
Electronic Location ID: e12897
Received 2021 Jun 4; Accepted 2022 Jan 16
Copyright: ©2022 Syed Sulaiman et al.
Copyright year: 2022
Copyright holder: Syed Sulaiman et al.
License: This is an open access article distributed under the terms of the Creative Commons Attribution License, which permits unrestricted use, distribution, reproduction and adaptation in any medium and for any purpose provided that it is properly attributed. For attribution, the original author(s), title, publication source (PeerJ) and either DOI or URL of the article must be cited.
License URL: https://creativecommons.org/licenses/by/4.0/

Keywords: Synovial fluid, Osteoarthritis, Rabbits, Surgical, Chemical, MALDI TOF/TOF mass spectrometry, Two-dimensional polyacrylamide gel electrophoresis, Proteomics

Funding: Fundamental Research Grant Scheme (FRGS) grant from the Ministry of Higher Education, Government of Malaysia FRGS/1/2018/STG03/UPM/02/3 The study was supported by the Fundamental Research Grant Scheme (FRGS) grant (FRGS/1/2018/STG03/UPM/02/3) from the Ministry of Higher Education, Government of Malaysia. The funders had no role in study design, data collection and analysis, decision to publish, or preparation of the manuscript.

==============================
Background

Animal models are significant for understanding human osteoarthritis (OA). This study compared the synovial fluid proteomics changes in surgical and chemical induced OA models.

Methods

Thirty rabbits either had anterior cruciate ligament transection (ACLT) procedure or injected intra-articularly with monosodium iodoacetate (MIA, 8 mg) into the right knee. The joints were anatomically assessed, and the synovial fluid proteins analyzed using two-dimensional polyacrylamide gel electrophoresis (2DGE) and MALDI TOF/TOF mass spectrometry analysis at 4, 8 and 12 weeks. The proteins’ upregulation and downregulation were compared with control healthy knees.

Results

Seven proteins (histidine-rich glycoprotein, beta-actin-like protein 2 isoform X1, retinol-binding protein-4, alpha-1-antiproteinase, gelsolin isoform, serotransferrin, immunoglobulin kappa-b4 chain-C-region) were significantly expressed by the surgical induction. They characterized cellular process (27%), organization of cellular components or biogenesis (27%), localization (27%) and biological regulation (18%), which related to synovitis, increased cellularity, and subsequently cartilage damage. Three proteins (apolipoprotein I-IV precursor, serpin peptidase inhibitor and haptoglobin precursor) were significantly modified by the chemical induction. They characterized stimulus responses (23%), immune responses (15%), biological regulations (15%), metabolism (15%), organization of cellular components or biogenesis (8%), cellular process (8%), biological adhesions (8%) and localization (8%), which related to chondrocytes glycolysis/death, neovascularization, subchondral bone necrosis/collapse and inflammation.

Conclusions

The surgical induced OA model showed a wider range of protein changes, which were most upregulated at week 12. The biological process proteins expressions showed the chemical induced joints had slower OA progression compared to surgical induced joints. The chemical induced OA joints showed early inflammatory changes, which later decreased.

Introduction

Osteoarthritis (OA) is a joint degeneration disease caused by the imbalance between cartilage degradation and synthesis, commonly due to aging. OA symptoms include joint pain, inflammation, morning stiffness, and decreased physical function, especially in the knee and hip. OA is characterized by deterioration of cartilage, formation of osteophyte and osteosclerosis, degradation of subchondral bone and tendons, inflammation of the synovium tissue, perforation of the trabecular plate, loss of trabecular bone and bone cysts (Burr & Gallant, 2012; Li et al., 2013).

OA includes cartilage disorder and subchondral bone deterioration (Burr & Gallant, 2012; Goldring & Goldring, 2016). Articular cartilage degradation involves overexpression of matrix catabolic enzymes, inflammation, oxidative stress, and deterioration of proteoglycan and collagen matrix. Around 9.6% of males and 18% of females exceeding 60 years of age are impacted by OA (World Health Organization, 2013), resulting in reduced quality of life and productivity as well as an increased economic burden or medical cost (Salter, Su & Lee, 2014). Current therapies, which include physiotherapy, orthopedic aids (orthoses), drugs, and surgery have limited efficacy. Pharmacological treatment has been limited due to adverse effects on cartilage (Huskisson et al., 1995) and gastrointestinal tract (Davies et al., 1997), whereas surgical intervention possesses many risks such as infection, blood clot and longer recovery rate.

Animal models used in pre-clinical OA research are classified into spontaneous and induced OA, further subcategorized into surgically or chemically induced OA models. Surgical OA induction modifies the joint strain/stress, leading to joint instability and damage. The anterior cruciate ligament transection (ACLT) is the most common method for surgical induced OA, entailing subchondral bone destruction and resulting in drastic cartilage changes (Lampropoulou-Adamidou et al., 2014). ACLT causes joint destabilization followed by post-traumatic osteoarthritis and reproduces the articular cartilage degradation after ACL injury (Piskin et al., 2007), or blunt trauma to cartilage, menisci, and subchondral bone (Teeple et al., 2013). The surgical model effectively yields a rapid OA progression; hence, it is convenient for short-term research but not for studies on early OA (Kuyinu et al., 2016). ACLT also causes changes in cell morphology and tissue equilibrium, especially in the femoral condyle cartilage (Buckwalter, Mankin & Grodzinsky, 2005). For chemical induced OA, monosodium iodoacetate (MIA) is most commonly used (Guingamp et al., 1997; Takahashi, Matsuzaki & Hoso, 2017). MIA suppresses the glyceraldehyde-3-dehydrogenase activity required for glycolysis in chondrocytes, resulting in apoptosis, cartilage degradation, proteoglycan loss, functional joint impairment and subchondral bone lesions (Pitcher, Sousa-Valente & Malcangio, 2016; Xie et al., 2012), similar to human OA.

Previous reports compared bone changes, histology, biochemistry and biomechanics of surgical versus chemical induced OA (Naveen et al., 2013) and concluded that chemical induced OA mimicked human OA. Each of the induced OA models has its own advantages, and no single model is considered the gold standard in assessing OA.

To the best of our knowledge, the synovial proteomic profiles between the two induced models have not been compared yet. This study analyses the proteome profiles of synovial fluids at different stages of surgical or chemical induced OA in rabbits, to provide further insights on the pathophysiology of OA and pave the road for future applications.

Methods

Animals

Thirty male New Zealand white rabbits (A-Sapphire Enterprise, Malaysia) aged 8–9 months and weighing 1.8–2.0 kg were placed in Animal Research Facility, Faculty of Veterinary Medicine, Universiti Putra Malaysia. Each rabbit was kept in an individual stainless-steel cage at 22–25 °C, under a 12hr/12hr light/dark cycle with 50–60% relative humidity. Rabbits were fed with commercial rabbit pellets (Penternakan Hong Lee Sdn. Bhd., Malaysia) and given fresh water ad libitum. The rabbits were acclimatized for one week prior to OA induction. Approval by the Institutional Animal Care and Use Committee (IACUC), Universiti Putra Malaysia was obtained for the experimental protocol (UPM/IACUC/AUP-R034). The rabbits were randomly grouped into surgically induced OA rabbits and chemically induced OA rabbits, with 15 individuals in each group. Rabbits were further subdivided into week 4, week 8 and week 12 time groups with five individuals in each group. OA was induced at the right knees of the animals, and the contralateral joint was used as a healthy control.

Preparation of animal model of surgically induced and chemically induced osteoarthritis

Anterior cruciate ligament transection (ACLT) was done for surgically induced OA group on the right stifle with a scalpel, under surgically sterile conditions, as described by Vignon et al. (1987). The animals were anesthetized with Zoletil® (Virbac, Milperra, Australia) at 3 mg/kg via intramuscular route and maintained with isoflurane (3%) (Piramal Healthcare, India). In order to expose the anterior cruciate ligament, medial patella dislocation was performed by lateral parapatellar arthrotomy approach with the stifle joint being fully flexed. Once visualized, irrigation with sterile saline was made onto the stifle joint. Subsequently, using 4-0 polydioxanone suture, closure of the joint capsule and subcutaneous tissue was done. Skin closure was done using 3-0 nylon suture. Tramadol (Duopharma, Kuala Lumpur, Malaysia) 2mg/kg was given twice daily for three days as painkiller. The animals were caged with unrestricted movement until anaesthetic recovery.

The OA chemical induction was administered with 0.32 ml of 25 mg/ml monosodium iodoacetate (MIA) (Sigma-Aldrich, Burlington, MA, USA) dissolved in saline, injected into the intraarticular space of the stifle joint, under general anaesthesia (Zoletil®) at 2 mg/kg via intramuscular route.

Euthanization and sample collection

Rabbits were visually inspected for any clinical signs, such as weight loss or immobility. In each group, rabbits were subdivided into week 4, week 8 and week 12 time groups. Because both joint amputations were required to collect the samples, rabbits were euthanized following the experiment with 120 mg/kg pentobarbital sodium (Vetoquinol, Magny-Vernois, France) to avoid suffering caused by risk of infection and also immobilization. Knee joints were removed and immediately fixed in 10% buffered formalin (Sigma-Aldrich). After 24 h, the muscle and tissue surrounding the femur and tibia were removed carefully to avoid damaging the cartilage surface. The proximal femur and distal tibia were placed in 10% buffered formalin. Medial and lateral femoral condyle and tibial plateau of left and right stifle joints were photographed by P7000 digital camera (Nikon, Tokyo, Japan) at macro setting. They were examined for gross morphological changes, including erosions at the medial and lateral articular cartilage surfaces. Synovial fluids were collected via arthrocentesis, placed in a 1.5 ml microcentrifuge tube and stored immediately at −80 °C until subsequent analysis.

Histology

Bones from harvested joint were decalcified immediately fixed in 10% buffered formalin (Sigma-Aldrich) and were decalcified with 10% formic acid (Nacalai Tesque, Japan) for 10 days. The femur and tibia were cut at a dorsal plane. Samples were dehydrated in a series of immersion in alcohol using Leica TP1020 Semi-enclosed Benchtop Tissue Processor (Leica Biosystem, Wetzlar, Germany). Next, the samples were embedded in paraffin using Leica EG1150H and EG11559 Modular Tissue Embedding Center (Leica Biosystems, Wetzlar, Germany) and sliced at 5 µm using Reichert-Jung 2045 Multicut Rotary Microtome (Leica Biosystems, Germany) and mounted on glass slides. Prior to staining, the slides were deparaffinized and hydrated using distilled water. Next, they were stained using Fast Green (FCF) solution for five minutes and rinsed with 1% acetic acid before being stained using 0.1% Safranin-O solution for five minutes. The slides were dehydrated and cleared with 95% ethyl alcohol and absolute ethyl alcohol, alternately at two minutes each for two times. The slides were then observed under microscope (Motic, China) under 20x magnification and scored independently by two blinded observers. The changes of articular cartilage were scored using OARSI Scoring System (Pritzker et al., 2006). Histological evaluation of subchondral bone was done by measuring the percentage of bone volume over tissue volume (BV/TV). The measurement was done according to Nagira et al. (2020). In brief, the compartments of osteochondral units were defined, including cancellous bone (Cn.B), bone marrow (BM) and trabecular bone (Tb.B)(Fig. 1). The calculation of BV/TV was done as follows: Bone volume (BV/TV, %) = (Cn.B region - BM regions)*/Cn.B region ×100, (*Cn.B region - BM regions = Tb.B regions) (Fig. 1). Calculations were done using ImageJ with bone histomorphometry method.

Figure 1 Definition of tissue compartment in subchondral bone.

White square dotted line, cancellous bone (Cn. B), yellow round dotted line, bone marrow (BM). Calculation of BV/TV: Bone volume (BV/TV, %) = (Cn.B region –BM regions)*/Cn.B region ×100, * Cn.B region –BM regions = Tb.B regions.

Statistical analysis

Sample size estimation using power analysis was performed before conducting the experiment, which was based on prior work that detected 23 significantly differentially expressed proteins between OA induction group and control group (Luo et al., 2018). A sample size of seven rabbits per group were determined (α = 0.05, power >80%). However, in view of the large sample size, sample size of 5 rabbits per group was chosen as suggested by the Animal Research Committee. Power analysis was done using G*Power software (Faul et al., 2007). Articular cartilage scoring data were tabulated as median and statistical comparisons for scoring between groups were analysed using the Kruskal-Wallis (K-W) non-parametric ANOVA and followed up with Dunn’s multiple comparison tests. BV/TV value of subchondral bone evaluation were analysed using one-factor analysis of variance (ANOVA) and multiple comparison among groups were analysed using Tukey’s HSD post hoc test. All analysis were done using Graphpad Prism 7 (Graphpad Software, San Diego, CA, USA). The level of significance was at α = 0.05.

Protein precipitation from synovial fluid

Proteins from synovial fluid were precipitated using modified TCA/Acetone precipitation method (Chen et al., 2005). Before protein precipitation, synovial fluid samples were centrifuged at 3,000×g at ambient temperature for 10 min to remove cells and cellular debris. The synovial fluid samples were pooled for each group and precipitated with cold 10% trichloroacetic acid (TCA) (Chemiz, Selangor, Malaysia) in 1:4 ratio inside a microcentrifuge tube. Samples were kept for 60 min in a −20 °C chiller, before centrifugation at 12,000×g for 60 min at 4 °C. The supernatant was removed without disrupting the protein pellet. Protein pellets were washed three times with cold 100% acetone and centrifuged at 14,000×g for 20 min, then the supernatant was removed. Finally, the microcentrifuge tube was inverted on the C-fold tissue and air-dried for 30 s.

Protein concentration estimation

Protein pellets were resuspended and solubilized with rehydration buffer containing 7 M urea, 2 M thiourea, 4% CHAPS, and 0.002% bromophenol blue. Protein concentration was determined using the 2D Quant Kit (GE Healthcare, Stockholm, Sweden), following manufacturer’s protocol. A standard curve was obtained using 2 mg/ml Bovine serum albumin (BSA) stock solution. 500 µl precipitant was inserted into each tube, vortexed and incubated at ambient temperature for 2–3 min, then 500 µl co-precipitant was added. Tubes were then vortexed and centrifuged for 5 min at 10 000 x g and supernatant discarded. The tubes were centrifuged again. The residual supernatant was decanted. Copper solution (100 µl) and distilled water (400 µl) were added to each tube, and precipitated protein was dissolved by vortexing. The working reagent (1 ml) was added to each tube and mixed by inversion instantaneously. Absorbance was measured using Cary®50 UV-Vis spectrometer at 480 nm (Varian Inc., Palo Alto, CA, USA).

Isoelectric focusing

Synovial fluid protein (250 µg) was rehydrated overnight using 13 cm immobilized pH gradient strips pH 3–10 ImmobilineTM DryStrip gels (IPG) (GE Healthcare, Stockholm, Sweden). The samples were focused with a low starting voltage followed by a voltage gradient of 15 000 to 20 000 V, with a 50 mA strip limiting current. Strips were stored at −80 °C until further analysis.

Two-dimensional gel electrophoresis (2DGE)

For 2DGE, strips were first equilibrated for 15 min with 50 mg dithiothreitol (DTT) in equilibrium solution (50 mM tris–HCl pH 8.8, 6 M urea, 30% glycerol, 2% SDS, 0.002% bromophenol blue) (GE Healthcare, Sweden), followed by 15 min alkylation using 125 mg iodoacetamide (IAA) (GE Healthcare, Sweden) in equilibrium solution. The strips were positioned on top of 12% gradient slab polyacrylamide gels and secured with molten 0.5% agarose in running buffer (25 mM tris pH 8.3; 198 mM glycine; 0.1 percent w/v SDS) with a drop of bromophenol blue. Gels were run at 140 V for 10 min, then at 200 V for 55–60 min using Ruby SE 600 system (GE Healthcare, Stockholm, Sweden). Gels were stained using colloidal Coomassie R350 (GE Healthcare, Stockholm, Sweden) and destained using destaining solution (40% methanol, 7% glacial acetic acid, 53% distilled water).

Gel images were obtained using a GS-800TM Optimized Densitometer for intra-sample and inter-sample variability evaluation (Bio-Rad Laboratories, Hercules, CA, USA). The digitized images were aligned and analyzed using Progenesis SameSpots software (Nonlinear Dynamics, Durham, NC, USA). According to manufacturer’s instruction, spots were detected, matched and normalized. Differential protein spots between control and induced models were determined using normalized spots and compared with reference gel image. Fold difference and p-values were calculated using one-way ANOVA. Threshold value was set at 1.5 fold change for upregulation and downregulation with p ≤ 0.05.

Protein digestion and matrix-assisted laser desorption/ionization time of flight/time of flight mass spectrometry (MALDI-TOF/TOF)

Gels were rinsed twice using distilled water, and 2DGE spots were excised manually under sterile conditions. Spots were digested in a two-day process. On day one, the excised gel spots were washed with 150 µl of 100 mM ammonium bicarbonate (ABC) for 10 min and vortexed. Following that, wash solution (50% acetonitrile in 100 mM ABC) was added twice, 5 min for each wash. This process was repeated until the gel dye was utterly destained. Reduction solution (150 µl of 10 mM DTT in 100 mM ABC) was then added and gel spots were incubated at 60 °C in water bath for 30 min. Alkylation solution (150 µl of 55 mM IAA in 100 mM ABC) was added before incubation in the dark for 20 min. The solution was removed, and wash solution was added twice, 20 min for each wash. Next, 50 µl of 100% acetonitrile was added and incubated at ambient temperature for 15 min. Gel spots were then vacuum-dried using ScanVac (Labogene, Denmark) for 1 h. Finally, 25 µl of 7 ng/µl trypsin solution were added, before incubation overnight at 30 °C. On the next day, 25 µl of 50% acetonitrile were added and gel spots were incubated for 15 min at ambient temperature. After centrifugation at 14,000×g, supernatant samples were collected and transferred to new tubes. Then, 20 µl of 100% acetonitrile were added before incubation for 15 min. The obtained solution was transferred into a new tube and the peptide was vacuum-dried and resuspended with 5 µl–10 µl of 0.1% trifluoroacetic acid. The digested protein sample underwent desalting and concentration using µZipTipC18 pipette tips (Millipore, Bedford, MA, USA). Protein samples were then analyzed using ultraflextreme MALDI-TOF/TOF mass spectrometer (Bruker, Bremen, Germany). Peptide-matrix compounds were loaded using the dried droplet technique on the instrument target plate. The used matrix compound was α-cyano-4-hydroxycinnamic acid (CHCA) (Bruker, Bremen, Germany). Samples were analyzed with MASCOT version 3.5 (Matrix science, USA) using the MASCOT search engine against the sequence of Oryctolagus cuniculus species in the NCBInr database. Samples with p ≤ 0.05 were accepted as significant.

Results

General condition of rabbit models

Compared to the chemically induced OA joints, the joint effusion development in the surgically induced OA joints showed a more severe time-dependent swelling and warmth. Both groups showed no clinical signs of pain, immobility or reduction in body weight and were healthy prior to euthanization. The rabbits’ average weight prior to euthanasia is 2.4 kg and ranged from 1.8 kg–3.6 kg.

Visual inspection of the joint and gross anatomy of the femoral condyle and tibial plateau

The gross morphology of the femoral condyle and tibial plateau showed no significant changes before week 4 and were similar to the control. The articular cartilage of the joints from both induced OA groups was smooth and polished without any surface irregularities. OA features such as surface erosions were observed on the femur and the tibia of both groups in week 8. Erosions were more notable on the femur lateral condyle cartilage but had generalized distribution on the tibia articular cartilage. The joints displayed similar significant osteoarthritic features in both induced OA groups at week 12. Joints from surgically induced OA had more apparent erosion on the medial condyle of femoral and tibial cartilages. In contrast, joints from chemically induced OA showed evenly distributed erosions at the articular cartilage surface.

Histopathological articular cartilage grading, subchondral bone evaluation and statistical analysis

Significant cartilage erosion was found in surgically induced and chemically induced OA group whereas control group showed a normal chondrocytes distribution and smooth cartilage surface in both femur and tibia. OARSI histopathology score was shown in Table 1 and Fig. 2, and representative histological images for femur and tibia was shown in Figs. 3 and 4. For femur, in the surgically induced group, uneven articular surface and superficial fibrillation were observed (Figs. 3B, 3C) at week 4 and week 8 after induction. Superficial zone fibrillation was observed during week 12. Chemically induced group showed a more severe articular cartilage changes indicated by cartilage erosion which was observed during week 8 and week 12 (Figs. 3G, 3H). For tibia, surgically induced group showed superficial fibrillation and chondrocytes apoptosis indicated by empty chondrons during week 8. During week 12, deep fibrillation and surface discontinuity was observed. Again, chemically induced group showed a more severe changes with surface erosion presence during week 8 and week 12 (Figs. 4G, 4H).

Table 1 OARSI histopathological grade of femur and tibia in the control, week 4, week 8 and week 12 for the surgically and chemically induced group.

	Control	Week 4	Week 8	Week 12	
	ACLT (median, range)	MIA (median, range)	ACLT (median, range)	MIA (median, range)	ACLT (median, range)	MIA (median, range)	ACLT (median, range)	MIA (median, range)	
Femur	0	0	0 (0–1)	0.5 (0–1.5)	1 (0.5–2)	1.5 (1–1.5)	1 (0–2.5)	4.25 (2–5.5)	
Tibia	0	0	0 (0)	1 (0.5–1)	0 (0–1)	3 (3–5)	2 (1.5–2.0)	5 (4.5–5.5)	

Figure 2 OARSI histopathological scores for (A) surgically induced femoral cartilage (B) chemically induced femoral cartilage (C) surgically induced tibial cartilage (D) chemically induced tibial cartilage Values are presented in median with 95% confidence interval (n = 5).

Significant difference (Dunn’s multiple comparisons, p < 0.05) between column with the same letter within the groups.

Figure 3 Representative histology images of femoral articular cartilage.

(A–D) Surgically induced group. (E–H) Chemically induced group. Safranin-O/fast green stained with 20×magnification. Surgically induced group showed apoptosis indicated by empty chondrons (red arrow) during week 4 and superficial fibrillation (red asterisk) during week 8. Chemically induced group showed surface erosion as early as week 8 (red arrowhead).

Figure 4 Representative histology images of tibial articular cartilage (A–D) surgically induced group. (E–H) chemically induced group. Safranin-O/fast green stained with 20x magnification.

During week 4, surgically induced group showed chondrocyte apoptosis (red arrow) and fibrillation (red asterisk) during week 8. Cartilage erosion (red arrowhead) were present during week 12. For chemically induced group, erosion can be observed during week 8 and week 12.

Histopathological evaluation of subchondral bone recorded decreased BV/TV value in surgically induced group for week 4 and week 8 in comparison with control (Fig. 5). At week 12, BV/TV value in surgically induced group increased slightly compared to week 8. For chemically induced group, BV/TV value decreased persistently from week 4 until week 12. The same pattern was recorded for both induction groups for tibia bone.

Figure 5 Subchondral bone BV/TV values for (A) femur and (B) tibia in week 4, week 8 and week 12 for the control, surgically (ACLT) and chemically (MIA) induced group.

No significant difference between groups were reported. The results are represented as mean (±SD).

Evaluation of protein yield and two-dimensional gel electrophoresis analysis

The precipitated protein concentrations from both induction groups ranged between 3.11–19.9 µg/ml. The 2DGE gels showed that throughout OA progression protein expression profiles are different from healthy control. The joints from surgically-induced OA displayed 28 significant matched spots (p ≤ 0.05; fold change expression ≥1.5) (Fig. 6A), whereas joints from chemical induced OA displayed a total of 23 matched protein spots (p ≤ 0.05; fold change expression ≥1.5) (Fig. 6B). The log normalized volume were calculated using SameSpot software. The spots were then manually excised and identified using MALDI-TOF/TOF.

Figure 6 Representative 2DGE (pI 3-10) image for rabbit synovial fluid in (A) surgically induced group and (B) chemically induced group after staining with colloidal Coomassie stain.

Gels from respective groups were scanned and analyzed with Progenesis SameSpot software and protein spots (circled in red) were selected to be picked.

Identification of proteins spots

Twenty eight and 11 non-redundant proteins were identified from joints with surgical induced and chemical induced OA, respectively. However, only seven proteins from surgical induced OA joints and three proteins from the chemical induced OA joints have significantly different (p ≤ 0.05) fold-change expressions (≥1.5) when analyzed as log normalized volume (Table 2).

Table 2 List of proteins and their log normalized volume for control, week 4, week 8 and week 12 for both induction groups.

The results are represented as mean (±SD).

Protein	Mean (± SD)	
	Control	Week 4	Week 8	Week 12	
Surgically-induced group					
alpha-1-antiproteinase F precursor	5.11(±0.3)	5.1(±0.37)	5.5(±0.3)	5.35(±0.23)	
histidine-rich glycoprotein	4.94 (±0.17)	5.03(±0.06)	5.1(±0.39)	5.41(±0.35)	
retinol-binding protein 4	5.26(±0.18)	5.49(±0.12)	5.12(±0.35)	5.38(±0.42)	
beta-actin-like protein 2 isoform X1	4.56(±0.28)	4.83(±0.21)	4.49(±0.38)	4.91(±0.04)	
Full=Ig kappa-b4 chain C region	6.19(±0.41)	6.35(±0.23)	6.22(±0.07)	6.38(±0.09)	
Serotransferrin	5.15(±0.2)	5.34(±0.3)	5.58(±0.51)	5.75(0.18)	
Gelsolin isoform	4.94 (±0.17)	5.03(±0.05)	5.1(±0.39)	5.41(±0.35)	
Chemically-induced group					
Serpin peptidase inhibitor	6.21 (±0.10)	6.47 (±0.08)	6.28 ± (0.22)	6.18 (±0.15)	
Haptoglobin precursor	4.85 (±0.04)	5.05 (±0.13)	5.38 (±0.15)	5.07 (±0.27)	
Apolipoprotein I-IV precursor	4.94 (±0.05)	5.45 (±0.20)	5.35 (±0.13)	5.21 (±0.11)	

The proteins were classified by their biological process functions (Table 3). In the surgically induced OA joints, proteins were characteristics of cellular process (27%), organization of cellular components or biogenesis (27%), localization (27%) and biological regulation (18%). For chemically induced OA joints, proteins were involved in stimulus responses (23%), immune responses (15%), biological regulations (15%), metabolism (15%), organization of cellular components or biogenesis (8%), cellular process (8%), biological adhesions (8%) and localization (8%).

Table 3 Characterization of proteins according to biological group for surgically and chemically induced group.

Biological process	Percentage (%)	
Surgically-induced group	
Cellular process	27	
Cellular Component Organization & Biogenesis	27	
Localization	27	
Biological Regulation	18	
Chemically-induced group	
Response to Stimulus	23	
Immune System Response	15	
Biological Regulation	15	
Metabolic Process	15	
Cellular Component Organization & Biogenesis	8	
Cellular Process	8	
Biological Adhesion	8	
Localization	8	

Discussion

The effusion developments in the surgically induced OA joints were observed as early as week 4 and increased significantly until week 12. Other reports on in vivo OA rabbit models detected severe joint effusion 2 weeks post-surgery, which grew prominent at week 8. The effusion in the surgically induced OA joints was more severe compared to the chemically induced OA joints, probably indicating greater post-operative inflammation severity (Bouchgua et al., 2009).

The gross morphology suggested that the OA damage in both groups was only manifest after 8 weeks post-induction. In surgically induced OA, joint lesions were more prominent on the medial condyle area, which is similar in humans, where the medial compartment is the most commonly affected. This relates to the meniscus functions, as the medial meniscus is the more important attachment for tibia movement at the knee (Campos et al., 2013). Thus, the ACLT model most effectively mimics human post-traumatic osteoarthritis (PTOA). Histological assessment of cartilage and subchondral bone is accepted as a gold standard in assessing and monitoring OA progression in animal models. This assessment can be used to observe different stages of joint degeneration (Rutgers et al., 2010).

Histological analysis revealed slight articular cartilage changes in surgically and chemically induced (Man & Mologhianu, 2014) OA during week 4 which indicated initiation of OA in both groups that may be pronounced over time. This is in line with previous report in which ACLT model in rabbits showed slower lesion progression between 3 and 6 weeks and the intensity of the lesion increases 6 weeks post-induction (Campos et al., 2013). During week 8 and week 12, more severe changes were observed in chemically induced group based on higher histological scoring. Similarly, previous study detected obvious degenerative changes in the cartilage as early as six to ten weeks after monosodium iodoacetate induction (Mohan et al., 2011). Subchondral bone histopathological evaluation in surgically induced group showed increased BV/TV values during week 12, suggesting the occurrence of bone remodeling indicating higher OA progression (Fell et al., 2019). As for chemically induced group, continuous decline in BV/TV throughout 12 weeks may be due to less loading of induced knee which triggered weight distribution alterations (Guingamp et al., 1997).

Induction using monosodium iodoacetate will lead to chondrocytes apoptosis that will first cause changes in morphological and histological features of articular cartilage before changes in subchondral bones (Naveen et al., 2013). On the other hand, surgical induction using ACLT method will cause destabilization of the joint and increased load at subchondral bone. Osteoclasts’ activity will increase and induced subchondral bone loss (Botter et al., 2009) and subsequently bone remodeling at later stages. Therefore, subchondral bone resorption is the initiating factor for articular cartilage degradation in surgically induced group and it can be concluded that changes in the bone preceded changes in the articular cartilage in ACLT induction (Yang et al., 2020).

The proteome profiles revealed four similar proteins detected in both groups: histidine-rich glycoprotein (HRG), serotransferrin, alpha-1-antiproteinase and immunoglobulin. Only HRG, serotransferin and alpha-1-antiproteinase expressions were significantly changed in the surgically induced OA joints. Immunoglobulin is one of the most highly expressed proteins, and its presence in both groups was expected since no depletion kit was used in this study. HRG, detected in both groups, is a multifunctional plasma protein interacting with the immune system. Previous studies showed that HRG binds strongly to some complementary proteins such as c4b binding protein, C1q, and factor H, and is one of the most present proteins in rheumatoid arthritis patients’ synovial fluid and plasma (Manderson et al., 2009). However, the role of HRG in OA is still unclear.

An upregulation of most identified proteins was observed in joints from surgically and chemically induced OA. Nevertheless, surgically induced OA displayed slight downregulation of alpha-1-antiproteinase protein, an acute-phase protein that protects tissue from uncontrolled proteolysis damage and is upregulated by pro-inflammatory interleukin-6 (Ucar et al., 2007). Such decrease indicates that inflammation and chondroprotection loss started at week 4 in the surgically induced OA model.

The retinol-binding protein 4 (RBP-4) was more upregulated in week 4 compared to week 8 and week 12. It belongs to the lipocalin family and is a blood retinol carrier positively correlated with the collagen degrading matrix metalloproteinases (MMPs) MMP-1 and MMP-13 (Scotece et al., 2018). MMP-13 was released in response to mechanical injury or joint destabilization (Rose & Kooyman, 2016), so this could explain RBP-4 expression in surgically induced OA joints.

In chemically induced OA joints, apolipoprotein I-IV precursor and serpin peptidase inhibitor proteins were most upregulated at week 4 compared to week 8 and week 12. Apolipoprotein is a pro-inflammatory acute-phase protein involved in lipid and cholesterol metabolism, found to be upregulated synovial fluids of OA patients (Nguyen et al., 2017). Apolipoprotein induces IL-6 expressions in human primary chondrocytes (De Seny et al., 2015). Serpin peptidase inhibitor inhibits serine proteases and MMPs regulation, aggrecanase plasmin, angiogenesis activity, tissue mitogens and inflammatory leukocyte proteases (Gobezie et al., 2007) in OA. These protein control processes such as inflammation and coagulation. The upregulation of apolipoprotein suggested that joint inflammation occurs early at week 4 post-induction. This consequently caused increased serpin peptidase inhibitor expression, which is negative feedback for MMP regulation following OA-related joint inflammation.

During week 8, downregulation of beta-actin-like protein-2 isoform X1 and RBP-4 was observed in surgical induced OA joints compared to the control group. Upregulation during week 4 followed by downregulation during week 8 indicated that OA progression was ongoing. Beta-actin-like protein 2 isoform X1 is a highly conserved protein responsible for cell motility, structure and integrity. Beta-actin-like protein 2 isoform X1 was most upregulated in the femur lateral secretome of early OA patients (Stenberg et al., 2013) and was also identified in synovial fluid of OA patients (Gobezie et al., 2007).

In chemically induced OA joints, the haptoglobin precursor was highly upregulated. Haptoglobin has immunomodulatory effects and reduces hemolysis-associated oxidative damage by binding with free hemoglobin (Cray, Zaias & Altman, 2009). The upregulation of haptoglobin precursor indicated that OA progressed from week 4 and persisted until week 8. A previous study reported haptoglobin expression was reduced as much as 70% in early shoulder OA patients (Wanner et al., 2013). The increased haptoglobin level in advanced OA is due to later upstream cytokine release (Liao et al., 2015).

In the surgically induced OA joints, the strongest upregulation of gelsolin and serotransferrin occurs at week 12, indicating advanced stage OA. Gelsolin is a major extracellular actin scavenging system component and a novel MMP-14 substrate, as shown in a previous study that demonstrates the cleavage of plasma gelsolin by MMP-14 (Park et al., 2006). MMP-14 can activate MMP-2 and MMP-13, involved in tissue turnover and upregulated in diseased state (Rose & Kooyman, 2016). Gelsolin was decreased in rheumatoid arthritis patients (Osborn et al., 2008) but increased in OA individuals (Mateos et al., 2012), where it is considered a putative biomarker. Thus, gelsolin upregulation in surgically induced OA joints from the early stage until week 12 indicates rapid disease progression because of MMP-14 activation.

Serotransferin upregulation was revealed in both types of OA joints. Serotransferin is secreted from the liver and involved in iron transport from absorption sites, from heme degradation to storage and utilization (Nylund et al., 2014). Serotransferin is a traditional acute-phase protein that reflects increased OA and RA inflammatory conditions. In this study, serotransferin levels are slightly downregulated in chemically induced joints and upregulated in surgically induced joints. Other studies reported serotransferin downregulation in OA (Gharbi, Deberg & Henrotin, 2011; Nylund et al., 2014), which contradicts the results from our surgically induced model. Serotransferrin was downregulated in early shoulder OA patients but upregulated by 50% in late OA patients (Wanner et al., 2013). This suggests that based on the protein expressed, surgically induced OA progression may be more advanced at week 12 compared to chemically induced OA.

Overall, surgically induced OA development occurred immediately after induction and continuously progressed until the end of the experimental duration (week 12), when most of the identified proteins were strongly upregulated. In chemically induced OA joints, the levels of the identified proteins were highest during the initial stage of induction and lowered at the final stage, suggesting that chemically induced OA develops rapidly during the initial stage but slows down or partially recovers at the later stages, based on the results at the molecular level. In the surgically induced group, destabilization of the joint will trigger the release of proinflammatory cytokines which subsequently activates acute phase response that were expressed by alpha-1-antiproteinase protein. Concurrently, retinol binding protein-4 and gelsolin release activated MMP related pathways. As for the chemically induced group, increased articular cartilage degradation will continue to release pro-inflammatory cytokines which in turn stimulated the release of acute phase proteins such as haptoglobin precursor and apolipoprotein I-IV precursor which involved in acute phase response. In addition, local response in the joint triggered a negative feedback mechanism on the inflammatory response through the expression of serpin peptidase inhibitor.

Surgically induced OA joints were associated with more cellular process changes. Synovitis is characterized by increased cellularity and cartilage damage at the advanced OA stages. Surgically induced OA mimics post-traumatic OA, which promotes angiogenesis (Liu et al., 2019). In surgically induced OA joints, angiogenesis involves cellular biological migration, growth and differentiation of endothelial cells, and organization of cellular components. Apolipoprotein is the metalloproteinase substrate that modulates angiogenic responses (Quintero-Fabián et al., 2019). Gelsolin, which is upregulated in surgically induced OA joints, is an important regulator of cellular functions for osteoclasts motility and podosomes (cell adhesion structures) formation (Silacci et al., 2004).

Chemically induced OA joints were associated with more diverse biological processes, although this group has fewer significant protein changes. The most represented proteins were involved in response to stimulus (23%) and in immune responses, biological regulation, and metabolic process (15%). The MIA induction disrupts chondrocytes glycolysis by suppressing glyceraldehyde-3-phosphatase dehydrogenase, subsequently causing mitochondrial pathway chondrocyte apoptosis involving reactive oxygen species (ROS) production (Jiang et al., 2013), neovascularization, necrosis and collapse of subchondral bone, as well as inflammation (Pitcher, Sousa-Valente & Malcangio, 2016). These processes involve changes in cell states and activities following a stimulus, thus causing major upregulation of stimulus–response proteins. In chemical induced OA joints, serpin peptidase inhibitor upregulation inactivates proteinase inhibition by ROS-associated oxidative inactivation, which causes further joint inflammation (Jones et al., 1998). Haptoglobin precursor and apolipoprotein are acute-phase proteins, released or activated in acute response following defective chondrocyte metabolism (Sipe, 1995). In auto-inflammatory and non-auto-inflammatory OA, immune regulation may be crucial for altering cell osmosis and inflammatory response in periostitis, which is associated with osteophytes formation in OA (Lu et al., 2014).

This study indicates that surgically induced OA developed more rapidly compared to chemically induced OA, based on the expressed biological process of proteins associated with advanced stage OA. The choice of OA induction is depending on the type of study that will be conducted, where surgically induced OA is suitable studies with a shorter time frame and chemically induced OA is suited for studies that needed a longer duration.

Several limitations were noted in this study. The main limitation to this study is the small numbers of animals in each group (n = 5) for each time point which also reflected the absence of sham control group. This is due to the suggestion by the Animal Care Committee as a consideration for the welfare of the rabbits. While this may hampers a more robust conclusion, it does provide significant preliminary data which may justify larger scale studies to be conducted. Also, decision to use contralateral joint as control was to reduce the number of animals to be sacrificed. In a study by Mustafy, Londono & Villemure (2018), contralateral joint can be accepted as a suitable control model as they showed high degree of similarities, with no significant differences among all evaluated mechanical, geometrical and morphological parameters. In addition, proteomics analysis was done using synovial fluid which was in direct contact with primary tissues of each joint. Hence, the results obtained for each joint would not completely correlated with each other as proteins expressed in each joint were relatively different as synovial fluid reflected the condition of each individual joint, showed focal changes in particular joints and therefore was reliable in characterizing changes in control and experimental joints.

The limitations in using rabbit models is the difference in joint biomechanics and gait in comparison with human, and structural difference in joint tissues (McCoy, 2015). However, although the translatability of each animal models to human clinical conditions are different, rabbit models are one of the widely used animal models as it is easily reproducible, low cost and easier in handling.

Changes in meniscus and synovium can also be observed to further elucidate disease progression as OA will cause degenerative changes in the meniscus by the loss of Type 1 and Type II collagen. In addition, synovial macrophage which is originated from synovium will be activated during inflammation process (Kuyinu et al., 2016). However, in this study, as synovial fluid is originated from synovium that releases various cytokines and pro-inflammatory mediators during disease progression (Danila, 2014), we would like to propose a non-invasive or less invasive method to assess the progression of OA using synovial fluid. This method will provide an early indication of OA development and intervention to stop or reduce further degeneration of joint and can represent OA progression.

Conclusions

This study showed the surgical induced model showed a wider range of proteome profile and had the highest upregulation of most proteins at week 12. On the other hand, chemically induced joints have slower OA progression compared to surgically induced joints, based on the biological process proteins expressed. The chemically induced joints showed inflammatory changes at the early phase but had decreased expression at the later stages.

Supplemental Information

Supplemental Information 1 Supplementary figures for Table 1

Spot quantification for proteins were analyzed by Progenesis SameSpot software. Spot volume normalization and calculation are performed by the software automatically.

Click here for additional data file.

Supplemental Information 2 Proteins identified and biological groups characterization

Click here for additional data file.

Supplemental Information 3 The ARRIVE guidelines 2.0: author checklist

Click here for additional data file.

We would like to thank Dr. Murshidah Mohd Asri and Dr. Goh Soon Heng for helping during the study.

Additional Information and Declarations

Competing Interests

Author Contributions

Animal Ethics

Data Availability

The authors declare there are no competing interests.

Sharifah Zakiah Syed Sulaiman conceived and designed the experiments, performed the experiments, analyzed the data, prepared figures and/or tables, authored or reviewed drafts of the paper, and approved the final draft.

Wei Miao Tan performed the experiments, analyzed the data, prepared figures and/or tables, authored or reviewed drafts of the paper, and approved the final draft.

Rozanaliza Radzi performed the experiments, authored or reviewed drafts of the paper, and approved the final draft.

Intan Nur Fatiha Shafie, Rozaihan Mansor, Suhaila Mohamed and Angela Min Hwei Ng analyzed the data, authored or reviewed drafts of the paper, and approved the final draft.

Mokrish Ajat and Seng Fong Lau conceived and designed the experiments, analyzed the data, authored or reviewed drafts of the paper, and approved the final draft.

Norasfaliza Rahmad performed the experiments, prepared figures and/or tables, authored or reviewed drafts of the paper, and approved the final draft.

The following information was supplied relating to ethical approvals (i.e., approving body and any reference numbers):

Approval by the Institutional Animal Care and Use Committee (IACUC), Universiti Putra Malaysia was obtained for the experimental protocol (UPM/IACUC/AUP-R034).

The following information was supplied regarding data availability:

The raw measurements are available in the Supplementary Files.

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
