# Peer review of "Synovial fluid proteome profile of surgical versus chemical induced osteoarthritis in rabbits"

_PeerJ, doi:10.7717/peerj.12897_

## Round 0.1 · original submission · Major Revisions

Although both reviewers found numerous issues and recommended rejection, I decided to give you a chance to reply to their critiques and to revise the manuscript. Please note that substantial additional research is needed, as many limitations cannot be substantially improved by altering the text.

Reviewer 1 ·

Basic reporting

I have reviewed the indicated manuscript which attempts to assess differences in the proteome of the synovial fluid (SF) derived from the knees of rabbits subjected to an ACL transection or exposed to a chemical toxin (monoiodoacetate).

While the results indicate some differences, the experimental approach has a number of severe limitations, and as well the outcomes. Several of the figures presented (Figures 1-3) are not very informative in their current form and could be deleted. In particular, Figure on is not relevant and Figure 2 does not indicate the state of the menisci, which is well known to be altered by 8 weeks post-ACL-T.

Aspects of the English language should be checked by an expert.

Experimental design

The experimental design has a number of flaws which are indicated below:
1. The ACL-T operation is an inflammatory stimuli and there is no sham group. While some of the inflammation may be transient, it can lead to changes in the SF.
2. Use of the contralateral joint may not be appropriate due to the contralateral effect that is well known.
3. The cells in the SF were not removed prior to freezing and thus, disrupted cells in the SF would contribute proteins.
4. The sex of the animals was not indicated-readily corrected!
5. The state of the synovium and the menisci was not indicated. In rabbits, after ACL-T, the medial meniscus undergoes degeneration by 8 weeks post-surgery. Also, the inflammed synovium contributes significant proteins to the SF including MMPs and other proteinases. 6. However, the MMPs, particularly MMP-13, were not apparently upregulated and this is well documented in the literature.
7. How did the authors decide on the dose of the chemical toxin to use? Is it a maximum to elicit a response. This toxin kills cells in more than just cartilage-synovium, menisci, capsule, ligaments. Were the affected tissues repopulated by 12 weeks? No real histology presented or performed apparently.

Validity of the findings

While the authors did detect some differences in the SF content following the two interventions, and the results may indicate differential responses, there are a number of limitations to the experimential design that limit interpretation and the significance of the differences. Based on other molecular changes that are known to occur following ACL-T in the female rabbit knee, it would appear that the primary methodolgical approach, that of 2D gels, is not sufficiently sensitive to detect the multitude of changes known to occur. The fact that the authors did not apparently remove the cells that are known to be in SF from both OA patients and the knees of preclinical animals with OA, presents a further complication to the interpretation.

Additional comments

While the concept of SF proteomics may reflect the on-going progression of OA in the knee, as well as the type of OA related to the "inciting" stimulus, there are a number of limitations to the experimental design of this study that cloud the interpretation of the findings.

Reviewer 2 ·

Basic reporting

nothing to add

Experimental design

In general the experimental design is well documented. However, the limited number of 5 animals per group is not discussed or accounted for. Based on the outcome what should be the number of animals needed to provide reliable outcome (what is the power of the study)?
Also the choice to have only a macroscopical assessment of the cartilage OA grade feels somewhat limited. Histological evaluation or even biochemical evaluation would be more appropriate, especially to place the findings in a broader context. Why is not the synovial inflammation included?
This could be helpful to put the proteins found and the validity of the models in more clinical relevant perspective.
What was the average amount of SF collected within each animal?

Validity of the findings

In general, the findings seems valid and are appropriate discussed. However, this reviewer is not certain what message the authors really want to bring across to the readers. What is the clinical context? The models differ in outcome, which can be expected but is the one more appropriate to use or does this depend on the intervention you want to apply in such a model in relation to the protein expressed.
Also the limitations of the study are limited presented, especially with regards the use of a rabbit model. for instance, the marked differences in joint biomechanics and gait when compared to humans. In addition to biomechanical differences, important structural differences also exist between rabbit and human joint tissues. Rabbit cartilage is *10 times thinner than human cartilage (0.3–0.7 vs 2–3 mm) but has much higher chondrocyte density. The distribution of cartilage zones is quite different between the 2 species, particularly because the thickness and cellularity of the transitional and radial zones are highly variable in the rabbit, even among sites within the same joint. The rabbit meniscus is also more cellular; it has less vascular penetration than the human meniscus; and it can heal rapidly. Rabbit cartilage has been reported to exhibit spontaneous healing, particularly in relativelyyoung animals. (some references: Pedersen DR, Goetz JE, Kurriger GL, et al. Comparative digital cartilage histology for human and common osteoarthritis models. Orthop Res Rev. 2013; 2013(5):13–20.; Poole R, Blake S, Buschmann M, et al. Recommendations for the use of pre- clinical models in the study and treatment of osteoarthritis. Osteoarthritis Car- tilage. 2010;18(suppl 3):S10–S16.; Cook JL, Hung CT, Kuroki K, et al. Animal models of cartilage repair. Bone
Joint Res. 2014;3(4):89–94.; Proffen BL, McElfresh M, Fleming BC, et al. A comparative anatomical study of the human knee and six animal species. Knee. 2012;19(4):493–499.)

Additional comments

In general this is a well written manuscript describing the synovial fluid proteome in two different induced rabbit OA models. In addition to the points addressed above is what message the authors really want to bring across? It seems somewhat vague formulated.

Minor:
abstract line 38: omit the word 'were'

---

## Round 0.2 · Major Revisions

Although the reviewer recommended rejection, I decided to give you another chance to revise your manuscript. Please address all the concerns of the reviewer and amend your manuscript accordingly.

Reviewer 2 ·

Basic reporting

In the revised manuscript there seems something wrong with the references. Several references are mentioned (e.g. Snickers, Mohan) in the text but are not listed.

Experimental design

The revised manuscript has been adapted to the comments of the reviewers, .

Validity of the findings

The limitations of the study have been adjusted but the limitations of the model used are not mentioned (e.g rabbit vs human), to provide a better context. Also the number of animals used could be better justified, the argument that other studies have used similar amounts of animals and found significant outcome is not a real valid argument, data can be still be coincidence. As suggested previously, a power calculation on the primary outcome of the present results can further support the validity of the present results obtained with the 5 animals.
Also the argument that the model reacts according other literature is a bit misleading as different species have been used (e.g rabbit vs dog Sniekers et al. 2008) which might lead to different results due differences in amount of loading of the joint etc.
The histology shows a bone-cartilage interface, so the authors are able to score the subchondral bone area (thickness, tightmark changes) in order to have a felling of the bone changes as is mentioned to be of importance by the authors in the discussion.

Additional comments

Though the authors try to improve the manuscript based on the comments of the reviewers, this reviewer feels that not all items have been addressed or answered completely.

There is no statement in the answers why not the menisci or synovial tissue has been included (or discarded) in the analysis while both reviewers make clear remarks on the importance of these tissues.
Or the use of the contralateral joint as a control or the lack of a sham is not discussed in the manuscript (part of the limitations).

Though the authors have stated there goal, this reviewer misses the discussions what this means in a more clinical or pathophysiological context? What do these results implicate? Surgical induction leads to a stronger proteomic response but to less structural damage in time as compared to the chemical induction which is clinical less relevant type of induction? This reviewer feels that this is missing in the discussion.

---

## Round 0.3 · accepted · Accept

It seems that critiques were addressed and the manuscript was revised accordingly.